# Lycopene, Mesoporous Silica Nanoparticles and Their Association: A Possible Alternative against Vulvovaginal Candidiasis?

**DOI:** 10.3390/molecules27238558

**Published:** 2022-12-05

**Authors:** Gabriela Corrêa Carvalho, Gabriel Davi Marena, Gabriela Ricci Leonardi, Rafael Miguel Sábio, Ione Corrêa, Marlus Chorilli, Tais Maria Bauab

**Affiliations:** 1School of Pharmaceutical Sciences, São Paulo State University (UNESP), Araraquara 14800-903, Brazil; 2Faculty of Medicine, University of Ribeirão Preto (UNAERP), Ribeirão Preto 14096-900, Brazil; 3Medical School, São Paulo State University (UNESP), Botucatu 18618-687, Brazil

**Keywords:** natural products, carotenoid, lycopene, mesoporous silica nanoparticles, antifungal activity, *Galleria mellonella*

## Abstract

Commonly found colonizing the human microbiota, *Candida albicans* is a microorganism known for its ability to cause infections, mainly in the vulvovaginal region known as vulvovaginal candidiasis (VVC). This pathology is, in fact, one of the main *C. albicans* clinical manifestations, changing from a colonizer to a pathogen. The increase in VVC cases and limited antifungal therapy make *C. albicans* an increasingly frequent risk in women’s lives, especially in immunocompromised patients, pregnant women and the elderly. Therefore, it is necessary to develop new therapeutic options, especially those involving natural products associated with nanotechnology, such as lycopene and mesoporous silica nanoparticles. From this perspective, this study sought to assess whether lycopene, mesoporous silica nanoparticles and their combination would be an attractive product for the treatment of this serious disease through microbiological in vitro tests and acute toxicity tests in an alternative in vivo model of *Galleria mellonella*. Although they did not show desirable antifungal activity for VVC therapy, the present study strongly encourages the use of mesoporous silica nanoparticles impregnated with lycopene for the treatment of other human pathologies, since the products evaluated here did not show toxicity in the in vivo test performed, being therefore, a topic to be further explored.

## 1. Introduction

Vulvovaginal candidiasis (VVC) is an opportunistic and endogenous fungal infection of the vulva and vagina, caused by *Candida* spp. [1,2,3]. This genus comprises of approximately 200 different species, being *Candida albicans* the responsible for 70 to 90% of the cases [4,5,6]. According to the United States Centers for Disease Control and Prevention (CDC), in the USA, VVC is considered the second most common cause of genital infection in women of reproductive age, and about 75% of women worldwide will have at least one episode of these disease in their lifetime [7,8]. Of these, 10 to 20% develop recurrent vaginal candidiasis, with four or more episodes per year [5,9].

From this perspective, the infection may be associated with recurrent and common habits among women, such as the use of tight and/or synthetic underwear, which make the environment favorable for *C. albicans* multiplication, in addition to the use of systemic or topical antibiotics, which reduces the protection of the vaginal flora, allowing *C. albicans* to colonize [4,10]. Furthermore, the etiologic agent infection can occur endogenously, once it is an opportunistic pathogen found in the healthy human microbiota, if the host presents an imbalance in its immune system, this fungus can spread and cause infections, or exogenously, through the contact with contaminated mucous membranes and secretions, or sexual contact, being more frequent in sexually active women, although it is not considered a sexually transmitted disease [11,12,13,14].

Clinically, VVC is characterized by intense vulvar itching, mainly by the presence of whitish discharge, by signs of inflammation accompanied by burning sensations and by occasional white-yellow spots on the vaginal walls and cervix [5,15]. Regarding treatment, despite the wide range of drugs available (such as azoles, polyenes, echinocandins and fluoropyrimidines), there are currently several reports in the literature of ineffective treatments due to susceptibility and resistance to various antifungals (mainly in the case of the most widely used class for the treatment of VVC, the azoles), making the study of new therapeutic options of fundamental importance, including substances of natural origin, such as lycopene [16,17,18,19,20,21,22]. A carotenoid responsible for the red color of fruits such as watermelon, pink grapefruit, papaya, red guava and especially tomatoes (main source of lycopene in the human diet) [23].

Among the various pharmacological properties of lycopene, the antifungal activity stands out, especially against *Candida* sp. However, it is worth noting that its insolubility in aqueous solvents is a limiting factor in therapy, as it results in low bioavailability. Therefore, lycopene impregnation in nanocarriers proves to be an interesting option for overcoming most of drawbacks concerning its application [20,23,24]. Mesoporous silica nanoparticles (MSN) comprise of an interesting nanoplatform for biomedical applications due to several properties including high thermal and chemical stability, adjustable pore structure and particle diameter as well as large surface area, free hydroxyls groups for functionalization, good biocompatibility, biodegradability, low toxicity and easy excretion by the body [25,26,27,28]. This type of nanoparticle, which was used for the first time for the purpose of controlled drug release in 2001, has been increasingly used in research addressing this purpose since it is capable of hosting different types of molecules through their interactions with the surface silanol groups of the silica matrix. It is worth mentioning that its porous characteristic is the primordial factor for the process of carrying and releasing molecules [29,30].

Although it is possible to find works in the literature addressing the use of both compounds for the treatment of *C. albicans* infections, their association gives to this work an innovative character, especially bearing in mind the incessant search for new, and promise, therapeutic options [23,25]. It is also worth noting that for the development of an ideal formulation, it is necessary, in addition to presenting excellent activity, to be considered safe specially in terms of toxicity, in this sense, the tests on *Galleria mellonella* larvae, although still scarce for the compounds mentioned here, are increasingly gaining space [31,32,33]. With this, the present work seeks to evaluate the suitability (in terms of in vitro antifungal activity and in vivo toxicity assessment) of using lycopene and MSN, isolated and in association, against VVC.

## 2. Results and Discussion

### 2.1. Characterization of Drug-Free MSNs and MSN@LYC and Drug Impregnation

By FTIR measurements, it was possible to observe in the spectra of both the MSN-R and MSN-C samples (Figure 1A,B, respectively) bands corresponding to symmetrical and asymmetrical axial stretching of Si-O-Si at 800 and 1085 nm respectively, characteristic of the silica matrix. Additionally, in the spectrum of Figure 1A, a band attributed to an axial Si-OH stretching around 960 nm is observed, characteristic of the silanols available on the surface and in MSN-R pores. A broad band around 3400 cm^−1^ was detected and attributed to a stretching of the O-H bond of water molecules. Finally, a band corresponding to symmetric angular deformation in the plane of the O-H bond was observed at 1630 cm^−1^, which was also attributed to the silanol groups available on the surface and pores of the MSN-R. Additionally, in the MSN-R spectrum it can also be observed that bands referring to the stretching of C-H (between 1400 and 1477 cm^−1^), indicative of organic matter residue [34,35,36,37]. In relation to Figure 1B, it is noteworthy that the MSN-C practically did not present OH groups while the MSN-R had this group preserved. The absence of the bands corresponding to the vibrations of the OH and Si-OH bonds for the MSN-C sample can be attributed to the thermal treatment of 550 °C carried out in the calcination process. The hydroxylation level of the matrices is extremely important for future tests of lycopene incorporation and release, very likely, different levels of hydroxylation will directly influence the drug incorporation and release processes.

Once characterized, we moved on to the impregnation stage, as can be seen in Table 1, the MSN-R presented slightly higher results than the MSN-C, therefore being the chosen one for the continuation of the work (called MSN@LYC).

Figure 2 shows MSN-R, free lycopene and MSN@LYC spectra. In the MSN@LYC spectrum, in addition to the NSM characteristic bands, it is possible to observe additional bands, such as the band in the region of 2850–2960 cm^−1^ corresponding to symmetric and asymmetric C-H stretching, a band that is also present in the LYC spectrum. It can also be observed that bands referring to the C-H stretching (between 1400 and 1477 cm^−1^) and the C=C stretching (1380 cm^−1^) are also present in both spectra. These three bands suggest the presence of organic matter in the pores, that is, it indicates that lycopene was properly incorporated into the MSN [38,39].

### 2.2. Evaluation of Antifungal Activity In Vitro by Determining the Minimum Inhibitory Concentration (MIC)

Table 2 presents the results obtained in the determination of MIC for the groups tested. MIC was determined as the minimum concentration capable of inhibiting more than 90% of fungal growth.

Although both strains are sensitive to amphotericin B (MIC ≤ 1.0 µg/mL), the ATCC 18804 strain presented a value below the lower limit recommended by CLSI [40] and by The European Committee on Antimicrobial Susceptibility Testing [41]. However, the result obtained agrees with a previous study carried out with the same strain, where a value of 0.06 µg/mL was also observed [42].

As recommended by CLSI (2020), the strain ATCC 18804 was sensitive to fluconazole since the value obtained was less than 2 µg/mL. The resistance profile of FMB-01 was confirmed since its MIC value was greater than ≥ 8 µg/mL [43].

As reported in the literature, lycopene showed antifungal activity against the tested strains of *C. albicans* [44,45]. However, the value obtained was higher than one of the few studies that reported lycopene MIC for a *C. albicans* strain, in this case TIMM 1768, where the authors obtained a value of 5 µg/mL [45]. It is worth mentioning that the *C. albicans* strain TIMM 1768 originates in the feces of patients with oral and gastrointestinal candidiasis, in contrast with ATCC 18804, used in this research, which was isolated from a cutaneous lesion caused by interdigital erosion [46,47]. On the other hand, Desai et al. [48] tested the activity of lycopene extracts, obtained from tomato and papaya, against *C. albicans* strain (ATCC 10231) and observed that they did not show activity at the concentrations tested, 5–100 µg/mL, which is in agreement with the obtained results, showing that at lower concentrations lycopene did not exhibit anti-*Candida* activity (7.81–250 µg/mL).

Despite being considered inert, both MSN showed activity against the *C. albicans* ATCC 18804 strain, and the MSN-C showed activity against the FMB-01 strain, this can be attributed to the fact that amorphous NSMs generate low to moderate levels of reactive oxygen species (ROS) [49]. Despite the fact that this production of ROS by MSN is not a consensus, it is known that its accumulation causes DNA damage and may even lead to apoptosis of cells such as *C. albicans*, which may explain the observed antifungal activity [44,50]. Finally, it is worth noting that both (MSN and lycopene) when in association did not show activity. What was not expected since there are reports in the literature where when protected within the nanocarrier the drug showed greater antifungal activity when compared to the free drug [42,51,52]. The issue of prolonged release may have been a limiting factor in this case, since small amounts of drug may be released gradually, which probably may not have followed the growth rate of the microorganism, in other words, the prolonged release here may have resulted in Insufficient amount of drug for the amount of fungus present on the microplate [53,54,55]. Another issue is the fact that the incorporated nanoparticles may have aggregated and sedimented at the bottom of the microplate well, which compromised its homogeneity in the well, thus limiting the drug release and action [56].

### 2.3. MFC Determination

Table 3 presents the results obtained in the determination of MFC for the groups tested.

As previously described in the literature, amphotericin B showed fungicidal activity [57]. Although fluconazole is considered fungistatic, in this study showed fungicidal activity in all tested strains [58,59]. However, it is worth mentioning that in the literature there are reports of fluconazole acting as a fungicide, mainly dose-dependent and in the VVC treatment [60,61,62].

The other groups tested, such as lycopene, MSN-C and MSN-R, showed fungistatic activity. It is known in the literature that lycopene has its antifungal action based on mitochondrial dysfunction (depolarization and cytochrome release) and ROS production (which can lead to cellular apoptosis) [44]. In addition, lycopene also acts destroying membrane integrity. It is worth mentioning that although lycopene interferes in budding division, it does not interfere in DNA replication [45]. Despite several studies in the literature elucidating lycopene action mechanism against *C. albicans* strains, studies addressing whether this activity would be fungicidal or fungistatic are scarce. This scarcity can also be observed for MSN. As aforementioned, the toxicity of MSN is still a controversial issue, but as in this study both nanoparticles showed activity against *C. albicans* ATCC 18804 strain and MSN-C against FMB-01, so it can be considered that, similar to lycopene, they act in ROS production [49,50,63].

The big difference between fungicidal and fungistatic compounds lies in the fact that the first one is able to inhibit 99.9% of fungal growth, while the second has a lower inhibition (50%). Another definition also used is that fungicidal compounds are those that inhibit the regrowth (for many hours) of the pathogen after a limited exposure for a certain period of time, whereas fungistatic compounds would be those where, after its removal, a rapid new growth would be observed [64,65]. Thus, it can be said that this classification may be related to the amount/time of cell death permanence and not to the mechanism of the death cause.

### 2.4. Toxicity Assessment in Galleria Mellonella Larvae

Despite the lack of studies evaluating the toxicity of lycopene and MSN through *G. mellonella* larvae, its use in predicting in vivo toxicity is of fundamental importance, since it fills the gap between in vitro and in vivo assays in mice, making a connection between them. It is also worth noting that this alternative in vivo model is increasingly gaining space due to advantages compared to those performed in mice, such as low cost (both operational and infrastructure), low biological risk and for being more ethically accepted [66,67]. Additionally, the immune system of these larvae is similar to mammals innate immune response, which makes their use even more desirable in predicting toxicity in humans [68]. As can be seen in Figure 3A,B, both MSN did not cause death or toxicity signs in the larvae, suggesting that their use can be considered safe at all doses tested.

Finally, the lycopene group (Figure 3C) showed 100% of viability at all evaluated concentrations, except for the highest one, 2000 mg/kg, where the observed viability was 93.3% in 48 h. However, it is worth noting that because this value is above the mean lethal dose (LD50) (necessary concentration of a certain substance to kill 50% of the larvae in 48 h) it can be suggested that lycopene does not present toxicity, being therefore suitable for therapeutic applications [67].

## 3. Material and Methods

### 3.1. Mesoporous Silica Nanoparticle (MSN) Synthesis:

The synthesis was performed according to NANDIYANTO et al. [69] and MATURI et al. [70] with adaptations. In a round bottom flask, 0.61 g of CTAB and 180 mL of distilled water were added and kept under stirring at 60 °C for 30 min. Posteriorly, 55.8 mL of octane, 105 μL of styrene (previously washed three times in 2.5 M NaOH), 0.14 g of L-Lysine, 6.42 mL of TEOS, 0.20 g of sodium dihydrochloride 2,2’- Azobis (2-methylproprylnamide) (AIBA), were added. Next, the reaction mixture was maintained by stirring at 60 °C for 3 h in nitrogen atmosphere. After this period, the suspension was transferred to a separatory funnel and kept overnight at room temperature for phase separation to occur. The supernatant was discarded, and the MSN suspension phase was centrifuged at 18,000 rpm for 15 min, then the precipitate was washed with ethanol and distilled water 5 times. After, the dried material was treated via reflux or calcination in order to remove organic residues.

#### 3.1.1. Reflux Treatment

The reflux was performed according to the methodology of Lungare, Hallan and Badhan [71] with adaptations. All material obtained in the synthesis was firstly dispersed in 60 mL of methanol, 1.5 mL of hydrochloric acid and 15 mL of chloroform, and then kept at reflux under stirring at 60 °C for 6 h. After this period, the suspension was centrifuged at 18,000 rpm for 15 min, then the precipitate was washed with methanol and distilled water 5 times. Lastly, the precipitate was dried and kept in a desiccator.

#### 3.1.2. Calcination Treatment

First, the porcelain crucible was calcined in a muffle furnace at 500 °C for 3 h and cooled in a desiccator [72]. Next, the material obtained in the synthesis was transferred to the porcelain crucible and calcined at a temperature of 550 °C for 6 h. Lastly, the material was stored in a desiccator [73,74].

### 3.2. Lycopene Impregnation in MSN

Firstly, the MSN were vacuumed for 1 h. Next, 5 mg of MSN were added in glass vials with lycopene solution in chloroform (2.5 mg/mL), which were kept under stirring at 100 rpm for 48 h under cooling (15 ± 5 °C). Subsequently, both lycopene-impregnated MSN (MSN@LYC) were centrifuged at 14000 rpm for 15 min and dried at room temperature under vacuum. In order to determine the amount of drug impregnated, the supernatant (previously filtered) was quantified by HPLC in a previously validated method. Next, the impregnation efficiency (IE)% and load capacity (LC)% was calculated according to Equations (1) and (2), respectively [23].
(1)$$IE\left(\%\right)=\frac{Total\;amount\;of\;drug\;added-Free\;amount\;in\;supernatant\;}{Total\;amount\;of\;drug\;added}\times 100$$
(2)$$LC\left(\%\right)=\frac{Encapsulated\;drug}{Nanoparticle\;weight}\times 100$$

### 3.3. Characterization of Drug-Free MSNs and MSN@LIC

The characteristic vibrations of the connections corresponding to the structure of the materials (MSN-R, MSN-C and MSN@LYC), that is, the formation of the silica network as well as the confirmation of the presence of the drug, were evaluated by vibrational absorption spectroscopy in the infrared region (FTIR). The FTIR spectra of the NSMs were obtained using an FTIR spectrometer, ALPHA Platinum ATR—FTIR from Bruker. The data were analyzed in the range of 4000 to 400 cm^−1^ with a resolution of 1 cm^−1^.

### 3.4. Evaluation of Antifungal Activity In Vitro by Determining the Minimum Inhibitory Concentration (MIC)

The sample preparation conditions were as follows: amphotericin B was solubilized in 20% DMSO and 80% PBS (32 µg/mL initial solution), fluconazole was solubilized in RPMI (256 µg/mL initial solution) and lycopene was solubilized in propanol, tween 80 and RPMI (10, 2 and 88%, respectively) at an initial concentration of 4000 µg/mL. MSN and MSN@LYC were solubilized in RPMI. MSN was maintained at a concentration equal to the MSN@LYC solution. MSN@LYC was maintained in an initial solution containing 4000 µg/mL of impregnated lycopene.

For this test, two *C. albicans* strains were used, the standard strain *C. albicans* ATCC 18804 and a clinical strain resistant to azoles provided by the Medical School of São Paulo State University (UNESP), Botucatu Campus (Certificate of Presentation of Ethical Assessment: 55758222.6.0000.5411), referred here as strain FMB-01. Prior to the test, the strains were kept in Sabouraud Dextrose Broth (SDB) with 20% glycerol at −20 °C. For performing the test, the strains were subcultured in 2 mL of SDB and incubated at 37 °C for 48 h. Subsequently, the cultures were transferred to SDB in order to prepare a suspension corresponding to 0.5 on the McFarland scale, which was confirmed by spectrophotometric reading at 530 nm (approximately 1 × 10^6^ cells/mL). The suspension underwent two consecutive dilutions, the first at 1:100 and the other at 1:20, in order to obtain a fungal suspension of 5.0 × 10^3^ cells/mL [75].

The test was conducted based on the Clinical and Laboratory Standards Institute (CLSI) document M27-A3 [75]. In a 96-well microplate, 100 µL of RPMI 1640 medium (pH 7.0–7.2) was added to all wells. Next, 100 µL of the test solutions (Lycopene and both MSN) were added in the wells of the first row of the plate and serial dilutions were performed in the 8 subsequent wells (each sample was applied to the first well of two columns to perform analysis in duplicate on each plate). Finally, 100 μL of the standardized suspensions of the microorganism were added to each well of the microplates. As a positive control, amphotericin B and fluconazole were used. Additionally, control of the culture medium, yeast growth, sterility of the test substances and solvents were carried out (it is worth noting that in these columns no microorganism was added).

As for concentrations, amphotericin B was tested in the range of 0.25–0.0018 µg/mL, fluconazole in the range of 64–1 µg/mL for ATCC and 1000–15.625 µg/mL for clinical strain and finally lycopene, calcined MSN (MSN-C) and refluxed MSN (MSN-R) in the range of 1000–7.81 µg/mL. The microplates were incubated at 37 °C for 48 h. The experiments were performed in triplicate. For plate reading, 20 µL of a freshly prepared 2% triphenyltetrazolium chloride solution was added to each well of the microplate and incubated at 37 °C for 2 h. Wells that lack microbial growth remain colorless and those that show growth turn pink.

### 3.5. Determination of the Minimum Fungicide Concentration (MFC)

After incubation of the 96-well microplates used in the MIC assay, the MFC determination was performed. With the aid of sterile wooden sticks, the mixture from each well (which showed antifungal activity) was transferred to Sabouraud Dextrose Agar plates, which were incubated at 37 °C for 48 h. MFC was defined as the minimum concentration necessary for the absence of colony forming units formation [76].

### 3.6. Evaluation of Toxicity in Galleria Mellonella Larvae

Sample preparation conditions: lycopene was solubilized in mineral oil (100%) at concentrations ranging from 2000 to 125 mg/kg. While MSN and MSN@LYC were solubilized in PBS (2000 to 125 mg/kg). *G. mellonella* larvae were kept in an environment with temperature (25 °C—regional average temperature) and controlled feeding [77,78,79,80,81,82]. Butterflies’ eggs were kept in a dark environment at 25 °C in large Petri dishes (140 mm, Sterilin) and maintained with artificial feeding. The artificial food was composed of a mixture of wheat germ, wheat flour, brewer’s yeast, powdered milk, honey and glycerol. The dry ingredients were mixed followed by addition of the wet ingredients. The diet was substituted twice a week and unused food was kept at 4 °C. All larvae were weighed and evaluated in a weight range of 0.2 to 0.3 g [83].

The in vivo toxicity determination test in *G. mellonella* larvae was carried out based on the study performed by Marena et al. [78] and Allegra et al. [84]. Seven groups were evaluated, namely orifice control (the larvae were only punctured), death (100% methanol), lycopene, lycopene solvent (100% mineral oil), MSN-C, MSN-R, and MSN solvent (phosphate buffer). In order to reach the proper weight for testing (0.2–0.3 g) the larvae were kept in containers with food in a Biochemical Oxygen Demand (BOD) incubator at room temperature. The sample dose applied to all groups, with a Hamilton Microliter^TM^ syringe, was 10 µL/larva. The application was performed on the penultimate proleg, as shown in Figure 4 Six concentrations were tested for the lycopene, MSN-C and MSN-R groups (2000, 1500, 1000, 500, 250 and 125 mg/kg). It is worth noting that for the orifice control, death (methanol 99.9%), lycopene solvent and MSN solvent groups, 10 larvae per group was used, while for the lycopene, MSN-C and MSN-R groups, an n of 60 larvae was used, 10 for each dilution.

After samples application the larvae were transferred to Petri dishes, and kept at 25 °C protected from light. The analysis of the acute toxic potential was performed by visually monitoring the larvae at 24, 48 and 72 h. In this monitoring, death (absence of reaction after physical stimulation), loss of motility, cocoon formation and melanization were evaluated. The analyses were performed in triplicate.

### 3.7. Statistical Analyzes

Statistical analyses were performed by bidirectional analysis of variance (ANOVA) one way followed by the Tukey post-test using the Graphpad Prism 7.0 software (Graphpad, San Diego, CA, USA.

## 4. Conclusions

The results suggest that both nanoplatforms, MSN-R and MSN-C, and lycopene can be used for the most diverse therapeutic applications, since they did not show toxicity in *G. mellonella* larvae. However, both components (MSN and Lycopene) when isolated showed only a slight activity against the tested strains. Additionally, when in association (MSN@LYC) this slight activity was not even observed. Finally, it is strongly encouraged that other applications of this association are evaluated since both the drug and the nanoparticle were non-toxic.

## Figures and Tables

**Figure 1 molecules-27-08558-f001:**
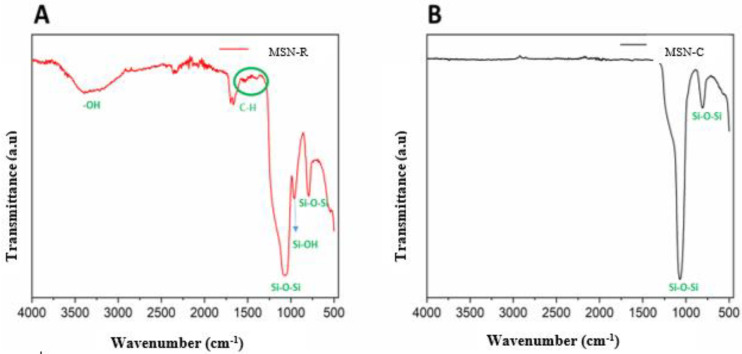
MSN-R and MSN-C infrared spectra. Legend: (**A**): MSN-R: refluxed mesoporous silica nanoparticle; (**B**): MSN-C: calcined mesoporous silica nanoparticle.

**Figure 2 molecules-27-08558-f002:**
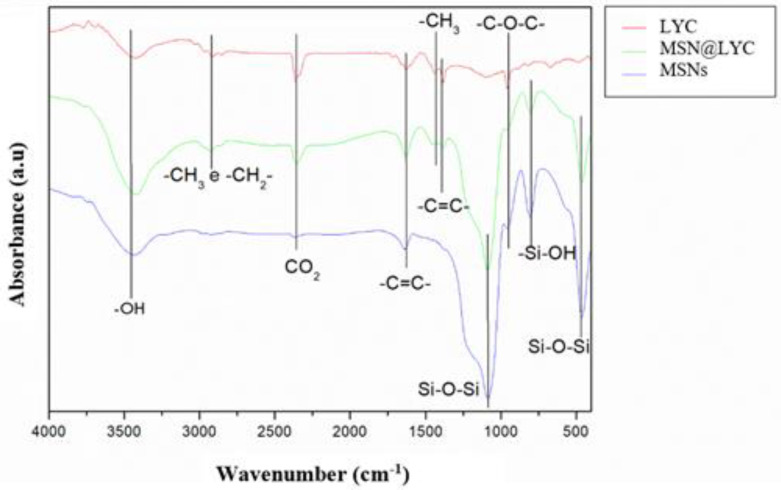
The infrared spectra of LYC, NSM and NSM@LYC diluted in KB pellet. Legend: LYC: free lycopene. MSN@LYC: mesoporous silica nanoparticle incorporated with lycopene. MSN: pure mesoporous silica nanoparticle.

**Figure 3 molecules-27-08558-f003:**
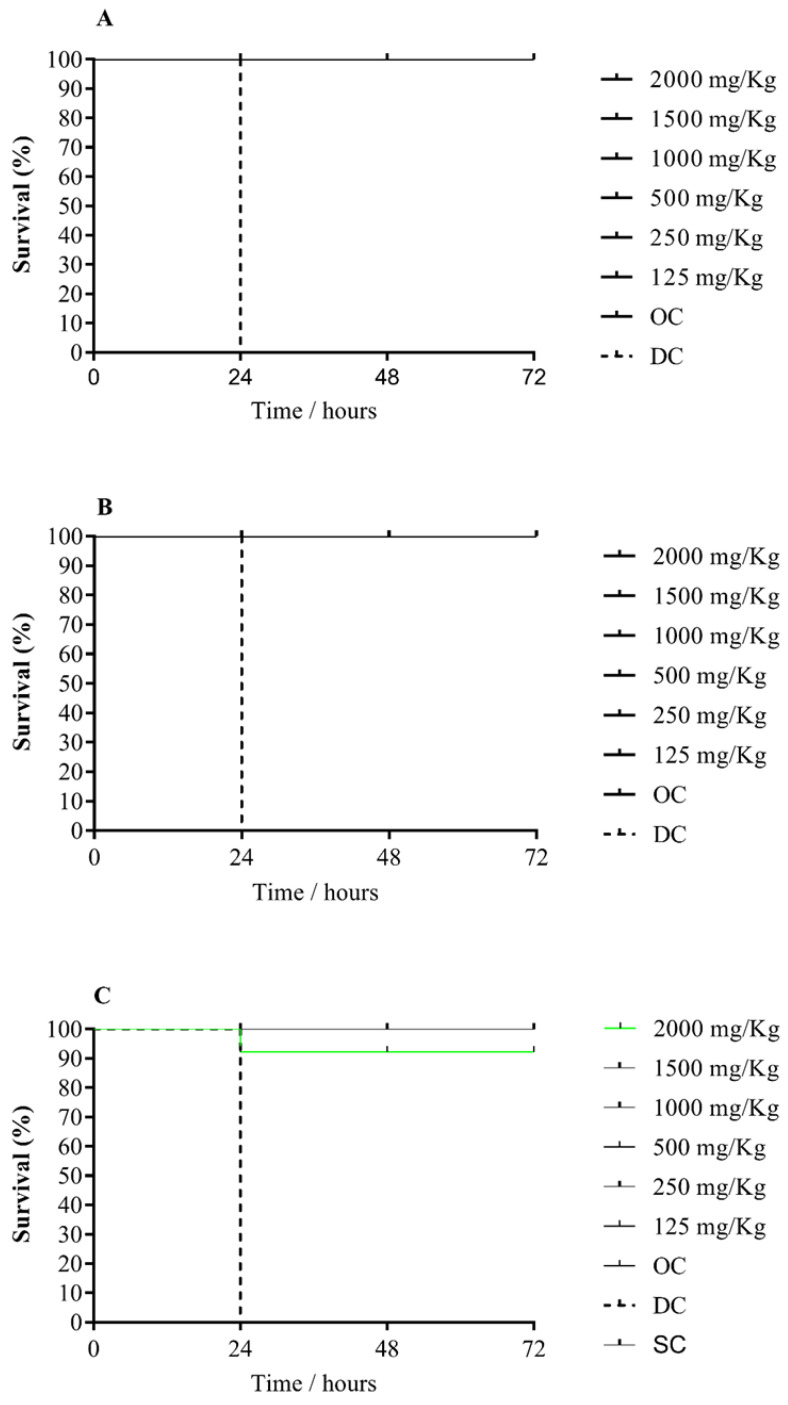
Kaplan-Meier survival curve of *G. mellonella* larvae for the analyzed groups at six different concentrations. Legend: (**A**): MSN-C. (**B**): MSN-R. (**C**): lycopene.. OC: orifice control group. DC: death control group. SC: solvent control group.

**Figure 4 molecules-27-08558-f004:**
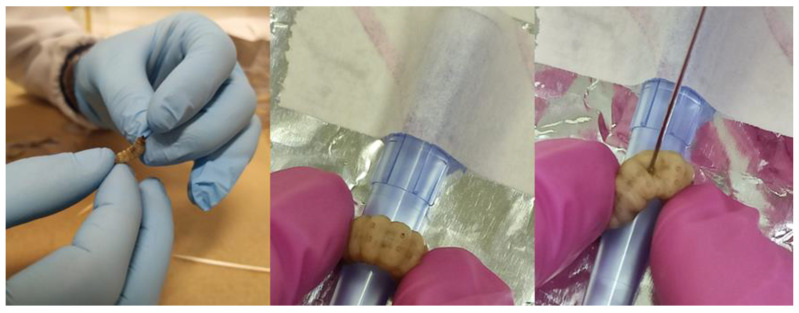
The sample application process in *Galleria mellonella* larvae.

**Table 1 molecules-27-08558-t001:** The values of impregnation efficiency, load capacity and amount of incorporated lycopene for both nanoparticles.

Sample	Impregnation Efficiency (%)	Load Capacity (%)	Amount of Impregnated Drug (g) in 5 mg of MSN
MSN-R	98.19 ± 0.12	49.10 ± 0.12	2.45
MSN-C	75.34 ± 0.22	37.67 ± 0.10	1.88

MSN: mesoporous silica nanoparticle. MSN-C: calcined mesoporous silica nanoparticle. MSN-R: refluxed mesoporous silica nanoparticle.

**Table 2 molecules-27-08558-t002:** The minimum inhibitory concentration of the groups tested.

Strains	Minimum Inhibitory Concentration of the Groups Tested (µg/mL)					
	Amphotericin B	Fluconazole	Lycopene	MSN-C	MSN-R	MSN@LYC
Clinical-azole resistant (FMB-01)	1.0	125.0	500.0	500.0	NA	NA
*C. albicans* ATCC 18804	0.06	1.0	500.0	500.0	1000.0	NA

Legend: NA: no activity in the tested range. MSN-C: calcined mesoporous silica nanoparticle. MSN-R: refluxed mesoporous silica nanoparticle. MSN@LYC: mesoporous silica nanoparticle impregnated with lycopene.

**Table 3 molecules-27-08558-t003:** The minimum fungicidal concentration of the groups tested.

Strains	Minimum Fungicidal Concentration (µg/mL)				
	Amphotericin B	Fluconazole	Lycopene	MSN-C	MSN-R
Clinical-azole resistant (FMB-01)	Fungicide at 1.0	Fungicide at 125.0	Fungistatic at 500.0	Fungistatic at 500.0	-
*C. albicans* ATCC 18804	Fungicide at 0.06	Fungicide at 1.0	Fungistatic at 500.0	Fungistatic at 500.0	Fungistatic at 1000.0

Legend: -: not tested. MSN-C: calcined mesoporous silica nanoparticle. MSN-R: refluxed mesoporous silica nanoparticle.

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
