# Peer review of "Lycopene, Mesoporous Silica Nanoparticles and Their Association: A Possible Alternative against Vulvovaginal Candidiasis?"

_molecules, 2022, doi:10.3390/molecules27238558_

Round 1

Reviewer 1 Report

In the manuscript molecules-2022261 by Carvalho et al. entitled ‘Lycopene, mesoporous silica nanoparticles and their 2 association: a possible alternative against vulvovaginal 3 candidiasis?‘ the authors have presented interesting work on an anti-fungal therapeutic agent. The text's language is very good, and very useful scientific discussions and conclusions have been shown in the text. These are a few comments that I have:

1.     It is unnecessary to talk too much about Vulvovaginal candidiasis. Please rewrite the introduction and also include more about mesoporous silica and lycopene.

2.     Where are the results and discussion for Mesoporous silica nanoparticle (MSN) synthesis, Reflux treatment, Calcination treatment, and Lycopene impregnation in MSN parts? How did you realize that the nanoparticles have been synthesized? How much IE and LC are?

Author Response

Manuscript Number: molecules-2022261

Title: LYCOPENE, MESOPOROUS SILICA NANOPARTICLES AND THEIR ASSOCIATION: A POSSIBLE ALTERNATIVE AGAINST VULVOVAGINAL CANDIDIASIS?

Prof. Dr. Maria Cristina Marcucci

Guest editor of Molecules

We are submitting the modifications related to the manuscripts assigned as molecules-2022261. We appreciate all the reviewers comments and suggestions, each of which was considered and answered. All new additional information or modifications are highlighted in red in the revised manuscript. We hope that this revised manuscript will be accepted for publication.

Remarks and suggestions from reviewers have been considered as follow:

Reviewer comments:

Reviewer: 1

Comments to the Author

In the manuscript molecules-2022261 by Carvalho et al. entitled ‘Lycopene, mesoporous silica nanoparticles and their 2 association: a possible alternative against vulvovaginal 3 candidiasis?‘ the authors have presented interesting work on an anti-fungal therapeutic agent. The text's language is very good, and very useful scientific discussions and conclusions have been shown in the text. These are a few comments that I have:

  1. It is unnecessary to talk too much about Vulvovaginal candidiasis. Please rewrite the introduction and also include more about mesoporous silica and lycopene.

Response: Adjustments have been made to the introduction (changes in red), thank you very much for your suggestion!

  1.   Where are the results and discussion for Mesoporous silica nanoparticle (MSN) synthesis, Reflux treatment, Calcination treatment, and Lycopene impregnation in MSN parts? How did you realize that the nanoparticles have been synthesized? How much IE and LC are?

Response: There really is a need for this information to fully understand the article, for this reason we have added two topics, "2.1) Characterization of drug-free MSNs and MSN@LYC and Drug impregnation" in the results and discussion section and "3.3) Characterization of drug-free MSNs and MSN@LIC" in the material and methodology section.

Reviewer 2 Report

Comments:

 The Authors should supplement the Material and Methods as there is no relevant information there, e.g.

1.   Where the used preparations, e.g. amphotericin B, fluconazole, lycopene, etc. came from, if the Authors prepared lycopene themselves, they should describe the procedure in detail. Please, complete the method with more details.

2.   How were the solutions of the preparations used? What was the solvent?

3.   There is no information on the caterpillars of the wax moth, where did they come from? Were they fed or starved during the experiment? I have objective about the temperature at which the larvae were kept. The optimal temperature for G. mellonella is 28-33°C, therefore only 25°C is a temperature that can cause a cold shock. This reaction strongly influences the immune response of the insects. Therefore, the results of the toxicity of the tested substances are not inconclusive / convincing, as the immunized insects were subjected to additional stresses.

4.   line 281, the Authors report that they used larvae weighing 0.2-0.4 g for the study. A two-fold difference in weight between individuals indicates that the insects were not at the same stage. Which already raises doubts as to the results obtained. It is known that the response of an insect to the same dose, but with such a difference in weight and simultaneously from different stages, will be different. Can the Authors explain how insects were selected in groups?

5.   The assessment of the toxicity of the preparation for only 3 days (72 hours) is insufficient, observation should be carried out until the appearance of adults (moths). As the death of insects was not tracked, the remaining parameters of the insect health index (rope 296) had to be additionally assessed. in addition to the health and mortality index, at least one immune response parameter should be assessed for toxicity assessments. The untreated insects and those treated with the solution in which the preparate was dissolved are appropriate controls for the toxicity test. And not the injection of 100% methanol, which would be toxic to everyone.

6.   Can the Authors explain why the toxicity of the substances used was checked for. If they differ from the preparations used, please provide details. If these are new molecules, they need to be characterized more closely.

7.   Authors suggest that lycopene, mesoporous silica nanoparticles (MSN) and their combination could be an alternative against vulvovaginal candidiasis. However, only experiments showing the effect in vitro were carried out. It would be interesting and important to show whether the preparations are effective against C. albicans in vivo. Galleria mellonella is a good model for this, because it can be kept at 37C typical for a human.

8.   The text should explain what the Authors refer to as the MIC. How is it possible that the MIC (Tab. 1) is the same as the MFC (Tab. 2). The only difference is that there is no MFC for MSN @ LYC.

9.   Please explain which MSN molecules were used to associate with lycopene. Were they synthesized according to 3.1 or MSN-C or MSN-R. Maybe the type of MSN used to associate with lycopene is important for the activity of the preparation?

10.    Materials and Methods lines 210 and 212 - please add methanol concentration

11.    Materials and Methods line 262  - no information on the concentrations of MSN @ LYC used, please complete

12.    There are no statistics

13.    Conclusion is a bit exaggerated based on the results obtained a suitable alternative to conventional approaches for surveillance and diagnostic purposes.

Author Response

Manuscript Number: molecules-2022261

Title: LYCOPENE, MESOPOROUS SILICA NANOPARTICLES AND THEIR ASSOCIATION: A POSSIBLE ALTERNATIVE AGAINST VULVOVAGINAL CANDIDIASIS?

Prof. Dr. Maria Cristina Marcucci

Guest editor of Molecules

We are submitting the modifications related to the manuscripts assigned as molecules-2022261. We appreciate all the reviewers comments and suggestions, each of which was considered and answered. All new additional information or modifications are highlighted in red in the revised manuscript. We hope that this revised manuscript will be accepted for publication.

Remarks and suggestions from reviewers have been considered as follow:

Reviewer comments:

Revisor 2: 

 The Authors should supplement the Material and Methods as there is no relevant information there, e.g.

  1. Where the used preparations, e.g. amphotericin B, fluconazole, lycopene, etc. came from, if the Authors prepared lycopene themselves, they should describe the procedure in detail. Please, complete the method with more details.

Response:  Dear reviewer, we added this information at the beginning of the topic 3.5 (in red).

2. How were the solutions of the preparations used? What was the solvent?

Response:  we added this information at the beginning of the topic 3.4 and 3.6 (in red).

3. There is no information on the caterpillars of the wax moth, where did they come from? Were they fed or starved during the experiment? I have objective about the temperature at which the larvae were kept. The optimal temperature for G. mellonella is 28-33°C, therefore only 25°C is a temperature that can cause a cold shock. This reaction strongly influences the immune response of the insects. Therefore, the results of the toxicity of the tested substances are not inconclusive / convincing, as the immunized insects were subjected to additional stresses.

Response: Dear reviewer, the maintenance method was added in item 3.6. Regarding temperature, studies report a longer lifespan of larvae at low temperatures of 15 to 18 ºC (Mowlds and Kavanagh, 2008). This is due to the fact that low temperatures result in a decrease in metabolism and an increase in the lifespan of the larvae. However, it is also reported in the literature that the larvae perfectly support higher temperatures of 25 to 37°C. Therefore, the G. mellonella model is considered suitable for analyzing the behavior of microorganisms and temperature toxicity analyzes in mammals (Pereira et al., 2018, Marena et al., 2022, Fuchs et al.,. 2010, Wodja et al., 2010, Cook and McArthur, 2010, Tsai et al., 2016).

4. line 281, the Authors report that they used larvae weighing 0.2-0.4 g for the study. A two-fold difference in weight between individuals indicates that the insects were not at the same stage. Which already raises doubts as to the results obtained. It is known that the response of an insect to the same dose, but with such a difference in weight and simultaneously from different stages, will be different. Can the Authors explain how insects were selected in groups?

Response: Dear reviewer, the selection of larvae in relation to weight was in line with the methodology found in the literature (Farosteiro et al., 2013, Scorzoni et al., 2013). The larvae selected for the experiments were from the same batch, avoiding experimental variability. However, we corrected the weight range, as the range used in this work was 0.2 to 0.3 mg (Trevijano-Contador et al., 2014). 

5. The assessment of the toxicity of the preparation for only 3 days (72 hours) is insufficient, observation should be carried out until the appearance of adults (moths). As the death of insects was not tracked, the remaining parameters of the insect health index (rope 296) had to be additionally assessed. in addition to the health and mortality index, at least one immune response parameter should be assessed for toxicity assessments. The untreated insects and those treated with the solution in which the preparate was dissolved are appropriate controls for the toxicity test. And not the injection of 100% methanol, which would be toxic to everyone.

Response: Dear reviewer, we performed the methodology as described in the literature. Studies indicate that the three-day method is important in determining acute toxicity (Allegra et al., 2018, Camargo et al., 2020). Our study indicates non-toxicity for three days, however, this does not prevent new tests with longer times.

6. Can the Authors explain why the toxicity of the substances used was checked for. If they differ from the preparations used, please provide details. If these are new molecules, they need to be characterized more closely.

Response: Dear reviewer, although there are studies reporting the toxicity of lycopene, they are done in cells. We did not find studies carried out on models of Galleria mellonella, which is an important factor. Furthermore, new studies can be carried out taking into account the results obtained in this work, such as, for example, the evaluation of the incorporation of lycopene in mesoporous silica nanoparticles, which would be unprecedented.

The use of alternative models in vivo has been increasingly desirable since there is a strong trend to apply the “3 Rs” rule (reduce, refine and replace) in experiments that intend to involve animal use.

7. Authors suggest that lycopene, mesoporous silica nanoparticles (MSN) and their combination could be an alternative against vulvovaginal candidiasis. However, only experiments showing the effect in vitro were carried out. It would be interesting and important to show whether the preparations are effective against C. albicans in vivo. Galleria mellonella is a good model for this, because it can be kept at 37 C typical for a human.

Response: Dear reviewer, thank you for the suggestion. Future tests may be performed in in vivo models.

8.The text should explain what the Authors refer to as the MIC. How is it possible that the MIC (Tab. 1) is the same as the MFC (Tab. 2). The only difference is that there is no MFC for MSN @ LYC. 

Response: MIC was determined as the minimum concentration capable of inhibiting more than 90% of fungal growth (we added this information to the manuscript). When the yeast is transferred under growth conditions, no growth is observed, that is, the concentration that inhibits (MIC) is capable of preventing growth when exposed to growth conditions, therefore, the MIC concentration is the same as the MCF.

9. Please explain which MSN molecules were used to associate with lycopene. Were they synthesized according to 3.1 or MSN-C or MSN-R. Maybe the type of MSN used to associate with lycopene is important for the activity of the preparation?

Response: There really is a need for this information to fully understand the article, for this reason we have added the topic, "2.1) Characterization of drug-free MSNs and MSN@LYC and Drug impregnation" in the results and discussion section.

10. Materials and Methods lines 210 and 212 - please add methanol concentration

Response: Dear reviewer, we added this information to the manuscript (in red)

11. Materials and Methods line 262  - no information on the concentrations of MSN @ LYC used, please complete

Response: 1000 - 7.81 µg/mL. We added this information to the manuscript (in red)

12.  There are no statistics

Response: . We added this information to the manuscript (item 3.7- in red)

13. Conclusion is a bit exaggerated based on the results obtained a suitable alternative to conventional approaches for surveillance and diagnostic purposes.

Response: Dear reviewer, we rephrase the conclusion

Round 2

Reviewer 1 Report

The authors have responded to my comments and made the necessary changes. Just please correct Figures XA and XB into Figure 1A and Figure 1B.